*Report*

EMBO
Molecular Medicine

# Peroxisome proliferator-activated receptor gamma (PPARγ) regulates lactase expression and activity in the gut

Mathurin Fumery[1,2,3,†], Silvia Speca[1,2,†], Audrey Langlois[1,2], Anne-Marie Davila[4], Caroline Dubuquoy[5], Marta Grauso[4], Anthony Martin Mena[1,2], Martin Figeac[6], Daniel Metzger[7], Christel Rousseaux[5], Jean-Frederic Colombel[8], Laurent Dubuquoy[1,2], Pierre Desreumaux[1,2,9] & Benjamin Bertin[1,2,*] iD

## Abstract

Lactase (LCT) deficiency affects approximately 75% of the world's adult population and may lead to lactose malabsorption and intolerance. Currently, the regulation of LCT gene expression remains poorly known. Peroxisome proliferator activator receptorγ (PPARγ) is a key player in carbohydrate metabolism. While the intestine is essential for carbohydrate digestion and absorption, the role of PPARγ in enterocyte metabolic functions has been poorly investigated. This study aims at characterizing PPARγ target genes involved in intestinal metabolic functions. In microarray analysis, the LCT gene was the most upregulated by PPARγ agonists in Caco-2 cells. We confirmed that PPARγ agonists were able to increase the expression and activity of LCT both *in vitro* and *in vivo* in the proximal small bowel of rodents. The functional response element activated by PPARγ was identified in the promoter of the human LCT gene. PPARγ modulation was able to improve symptoms induced by lactose-enriched diet in weaned rats. Our results demonstrate that PPARγ regulates LCT expression, and suggest that modulating intestinal PPARγ activity might constitute a new therapeutic strategy for lactose malabsorption.

**Keywords** hypolactasia; intestinal epithelial cells; lactase; lactose intolerance; PPARgamma

**Subject Categories** Metabolism; Pharmacology & Drug Discovery

## Introduction

Peroxisome proliferator-activated receptor gamma is a member of the nuclear receptor superfamily of ligand-activated transcriptional factors and a master gene for the control of glucose homeostasis and lipid metabolism (Tontonoz & Spiegelman, 2008). To date, most studies have evaluated the role of PPARγ in major metabolic organs such as liver, adipocytes, pancreas, or skeletal muscles (Ahmadian *et al*, 2013), leading to target PPARγ for the treatment of type 2 diabetes with the development of the thiazolidinedione (TZD) class of drugs (Lehmann *et al*, 1995). Beside adipocytes, the other major tissue expressing PPARγ is the intestine (Fajas *et al*, 1997). Genomic profiling of intestinal epithelial cells (IEC) stimulated with PPARγ agonists, as well as functional studies in mice, started to reveal the roles played by this receptor in the gut (Bertin *et al*, 2013). However, its precise functions within the intestine are poorly known and most of its target genes, notably in IEC, remain to be characterized.

The aim of this study was therefore to identify PPARγ target genes involved in IEC metabolic functions and homeostasis. Our results identified the gene encoding lactase (LCT) enzyme as a new gene regulated by PPARγ in IEC. Our data demonstrated that both synthetic and natural PPARγ agonists are able to increase the expression and activity of LCT *in vitro* and *in vivo*. The PPARγ ligand-dependent improvement of symptoms induced by a lactose-enriched diet in rodent further supports the important role played by the activation of PPARγ in lactose metabolism.

1 U995—LIRIC—Lille Inflammation Research International Center, Univ. Lille, Lille, France
2 Inserm, U995, Lille, France
3 Service d'Hépatogastroentérologie, Centre Hospitalier Universitaire d'Amiens, Université de Picardie Jules Verne, Amiens, France
4 UMR0914, Institut National de la Recherche Agronomique/AgroParisTech, Université Paris-Saclay, Paris, France
5 Intestinal Biotech Development, Lille, France
6 Functional and Structural Genomic Platform, Université de Lille, Lille, France
7 Institut de Génétique et de Biologie Moléculaire et Cellulaire, CNRS, UMR7104/INSERM U964/Université de Strasbourg, Collège de France, Paris, France
8 The Henry D. Janowitz Division of Gastroenterology, Icahn School of Medicine at Mount Sinai, New York, NY, USA
9 CHU Lille, Service des Maladies de l'Appareil Digestif et de la Nutrition, Hôpital Claude Huriez, Lille, France
*Corresponding author. Tel: +33 3 2062 7738; E-mail: benjamin.bertin-2@univ-lille2.fr
†These authors contributed equally to this work

# Results and Discussion

## Lactase mRNA, protein, and activity are induced by PPARγ agonists in Caco-2 cells

Gene expression profiles of the Caco-2 intestinal epithelial cell line stimulated by PPARγ agonists were first assessed by microarray analysis. We used three different PPARγ agonist: the well-characterized pioglitazone (Pio; 1 μM) belonging to the TZD drug class (Momose *et al*, 1991), 5-amino salicylic acid (5-ASA, 30 mM) (Rousseaux *et al*, 2005), and a new PPARγ modulator we developed and named GED-0507-34-Levo (GED; 1 and 30 mM) (Pirat *et al*, 2012; Mastrofrancesco *et al*, 2014). Among the 44,000 genes tested, we observed that the LCT gene was the most upregulated gene in cells treated with Pio and GED. The LCT gene was significantly 5.28-fold ($\pm$ 0.55; $P < 0.05$) upregulated by 1 mM GED, 8.28-fold ($\pm$ 1.7; $P < 0.05$) upregulated by 30 mM GED and 17.93-fold ($\pm$ 5.1; $P < 0.05$) upregulated for Pio compared to unstimulated cells. 5-ASA also upregulated LCT mRNA expression to the same extend (8.76-fold $\pm$ 2.06, $P < 0.05$) (GEO Series accession number GSE68852; http://www.ncbi.nlm.nih.gov/geo/query/acc.cgi?acc=GSE68852). These results were confirmed in independent experiments by evaluating LCT gene expression by quantitative RT–PCR (qRT–PCR). Significant LCT overexpression was induced by 1 mM GED (5.76 $\pm$ 0.89-fold change), 1 μM Pio (14.77 $\pm$ 1.37-fold change), and 30 mM 5-ASA (9.57 $\pm$ 1.96-fold change) (Fig 1A). In order to strengthen these results, we also evaluated the ability of two other PPARγ agonists to increase LCT expression: rosiglitazone (Rosi, 1 μM), another TZD drug class ligand, and the trans-10, cis-12-conjugated linoleic acid (CLA, 1 mM) isomer, a natural PPARγ modulator. Both of them significantly increased LCT mRNA expression (Rosi, 12.08 $\pm$ 2.00-fold change; CLA 4.03 $\pm$ 0.27-fold change) (Fig 1A). Dose–response evaluation showed that LCT mRNA upregulation was optimal with 1 mM GED, 1 μM Pio, 1 mM CLA, and 10 μM Rosi in Caco-2 cells (Fig EV1). In addition, immunostaining and Western blot analysis demonstrated induction of LCT protein expression levels in Caco-2 cells stimulated by GED, Pio, Rosi, and CLA (Fig 1B and C). We then evaluated the potential induction of LCT activity in Caco-2 cells by PPARγ agonists by measuring the rate of glucose production in culture supernatant resulting from the action of LCT after incubation of a monolayer of Caco-2 cells with lactose (Dahlqvist, 1968). Stimulation of Caco-2 cells by 1 mM GED, 1 μM Pio, 1 μM Rosi, and 1 mM CLA significantly increased LCT activity compared to untreated cells (from more than twofold and up to more than ninefold) (Fig 1D), without modification of glucose

uptake of Caco-2 cells (Appendix Fig S1). This PPARγ effect on LCT gene induction was not extended to other disaccharidases expressed by Caco-2 cells such as sucrase-isomaltase (SIM) and maltase-glucoamylase (MGAM) (Fig EV2).

Several elements of the genetic control of LCT gene expression have been elucidated (Troelsen, 2005; Curry, 2013). Among the single nucleotide polymorphisms characterized in the human LCT gene, two major polymorphisms, $C/T_{13910}$ and $G/A_{22018}$, were linked to hypolactasia (LCT gene expression deficiency) with homozygous $CC_{13910}$ and $GG_{22018}$ genotypes associated with the lactase non-persistent phenotype (Enattah *et al*, 2002; Swallow, 2003; Troelsen, 2005). The functional link between the C(–13910) allele and epigenetic changes that lead to lactase non-persistence has been recently established (Labrie *et al*, 2016). Interestingly, Caco-2 cells were found to be $CC_{13910}$ and $GG_{22018}$ (Fig EV3), suggesting that PPARγ agonists may be able to control LCT expression in lactase non-persistent genotypes.

Altogether, these data demonstrate that PPARγ agonists are able to induce LCT mRNA and activity in Caco-2 cells which possess the hypolactasia-associated genotype.

## PPARγ is a transcriptional regulator of the LCT gene

These results led us to investigate the presence of PPAR response element (PPRE) sequences in the LCT gene promoter. *In silico* analysis of the 3,000 base pairs upstream from the transcription start site of the human LCT gene revealed the presence of several potential PPRE, DR1, and DR2, by which PPARγ may regulate LCT gene expression (Fig 2A and Appendix Fig S2). Chromatin immunoprecipitation analysis of these putative PPRE revealed notably that a DR2 located between −223 bases pairs (bp) and −210 bp upstream of the transcription start site was bound by PPARγ within the LCT gene promoter in Caco-2 cells stimulated for 24 h by 1 mM GED. Quantitative PCR analysis showed a 2-fold increase of the amount of PPARγ bound to this PPRE after 1 mM GED stimulation compared to unstimulated cells (Fig 2A). A genomic fragment containing this DR2 was cloned upstream to the luciferase gene into a pGL4 vector (pGL4Luc-promLCT construct) and tested in a reporter gene assay in Caco-2 cells. In these transfected cells, luciferase activity was significantly increased after GED stimulation compared to untreated cells (Fig 2B). Similar results were obtained with pioglitazone (Appendix Fig S3). In order to confirm the involvement of the DR2 as a functional response element, we modified the sequence of the response element by site-directed mutagenesis in the pGL4Luc-promLCT reporter construct. We created two

---

**Figure 1.  PPARγ agonists specifically induce LCT expression and activity in Caco-2 cells.**

A   Quantitative PCR (qPCR) analysis of LCT gene expression in stimulated Caco-2 cells. Cells were stimulated for 24 h with each agonist. Results represent the fold change of LCT gene expression normalized to GAPDH level. The expression level measured in control cells was used as reference and defined as 1.

B   Immunofluorescence staining of Caco-2 cells for LCT protein (green). Cells were stimulated for 24 h with each agonist. Nuclei are stained with DAPI (blue). Non-relevant IgG was used as control ("IgG control"). Scale bar, 100 μm. Magnification ×20.

C   Western blot analysis of LCT protein expression from stimulated Caco-2 cells. Densitometric analysis was used to quantify LCT protein.

D   LCT activity in Caco-2 cells stimulated for 24 h. Results represent the fold change of LCT activity with respect to the activity measured in control cells arbitrarily defined as 1.

Data information: (A, D) Data are expressed as mean $\pm$ SEM (two to four independent experiments). Statistical analysis: two-tailed nonparametric Mann–Whitney $U$-test. ***$P < 0.0001$.

Source data are available online for this figure.

## A    LCT gene expression (mRNA)

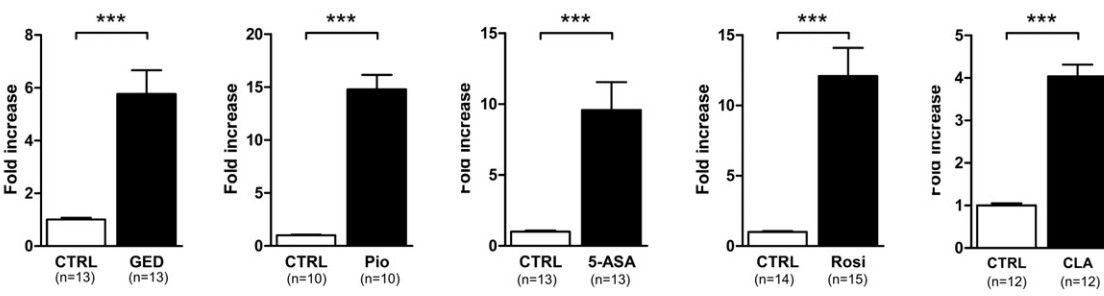

## B    LCT protein expression (immunofluorescence)

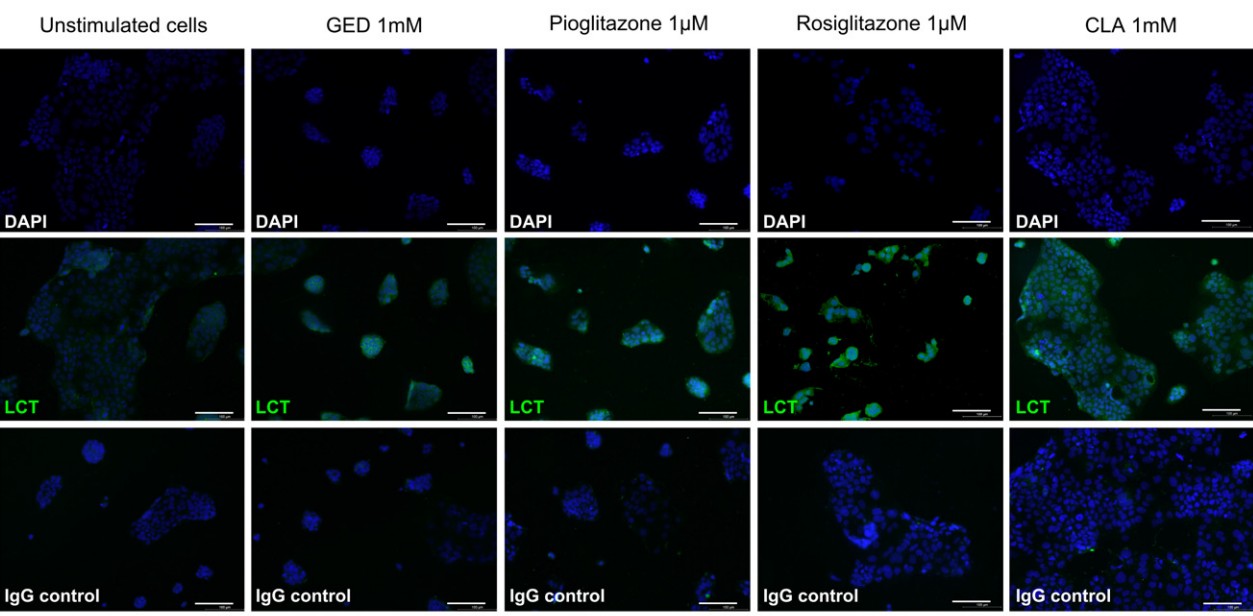

## C    LCT protein expression (Western-Blot)

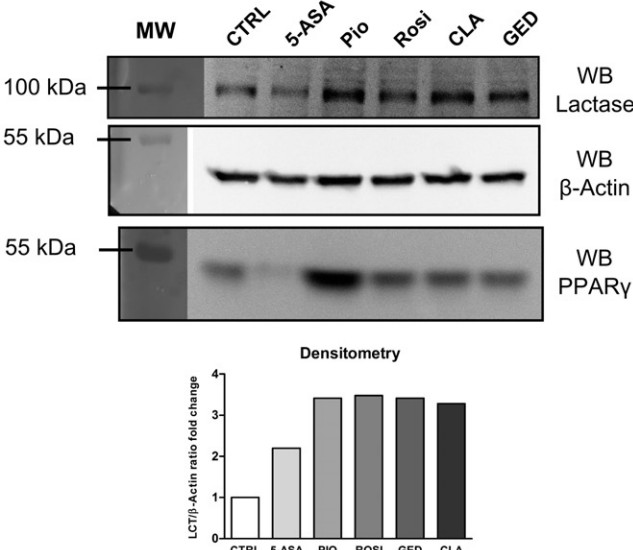

## D    LCT activity

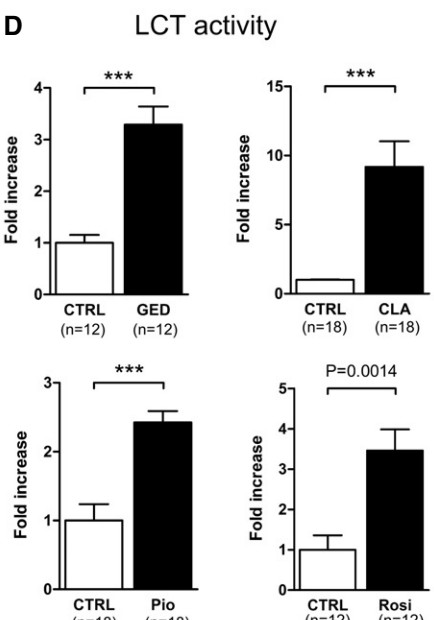

Figure 1.

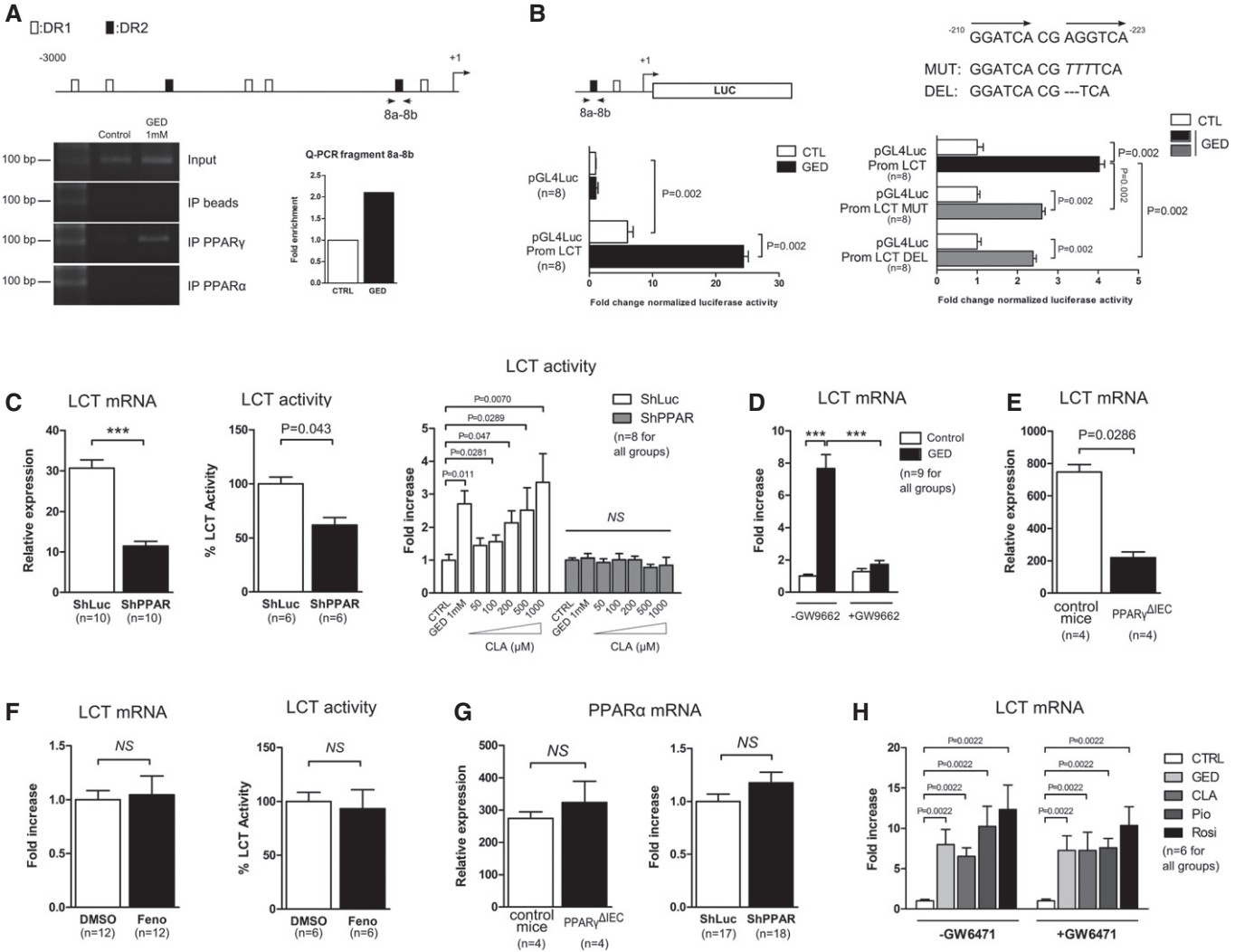

**Figure 2. PPARγ is a transcriptional regulator of the LCT gene.**

A   Chromatin immunoprecipitation (ChIP) assay. The top diagram depicts the PPRE predicted by *in silico* analysis. The picture shows PCR amplification of the 8a–8b fragment in ChIP assay from Caco-2 cells. Graph bars represent quantification of the 8a–8b fragment by qPCR. Results are expressed as fold enrichment with amplification from control cells defined as 1.

B   Luciferase gene reporter assay in Caco-2 cells transfected with pGL4Luc-PromLCT, pGL4Luc-PromLCT MUT, and pGL4Luc-PromLCT DEL reporter constructs. Cells transfected with empty pGL4Luc were used as control. Results represent the fold change luciferase activity normalized for protein content.

C   LCT gene expression measured by qPCR and LCT activity in PPARγ knockdown Caco-2 cells (ShPPAR) compared to control cells (ShLuc). LCT activity in ShPPAR cells compared to ShLuc cells stimulated by GED and CLA. The activity levels measured in control cells were arbitrarily defined as one.

D   Effect of GW9662 on GED-dependent induction of LCT gene expression in Caco-2 cells. LCT gene expression was determined by qPCR. Results represent the fold change of LCT gene expression. The expression level measured in control cells (w/o GED and GW9662) was used as reference and defined as 1.

E   LCT mRNA expression in the proximal small intestine of PPARγ^ΔIEC mice. Results represent the mean ± SD.

F   LCT mRNA expression and activity in Caco-2 cells stimulated with fenofibrate compared to control cells (DMSO).

G   PPARα mRNA expression in the small intestine of PPARγ^ΔIEC mice (left) and in Caco-2 ShPPARγ/ShLuc cells (right). For mice results, data represent the mean ± SD.

H   Effect of GW6471 on GED-, CLA-, Pio-, and Rosi-dependent induction of LCT gene expression in Caco-2 cells. Results represent the fold change of LCT gene expression. The expression level measured in control cells was used as reference and defined as 1.

Data information: Data are expressed as mean ± SEM (two to four independent experiments) (except for panels A, E and mouse data shown in G). Statistical analysis: two-tailed nonparametric Mann–Whitney *U*-test. ***$P < 0.0001$; *NS*, not significant.

new reporter plasmids: In the first one, the "AGG" sequence within the DR2 was mutated into "TTT" (pGL4Luc-promLCT MUT construct), and in the second one, the "AGG" sequence was entirely deleted (pGL4Luc-promLCT DEL construct) (Fig 2B). Although these two constructs were still responsive to GED in transient transfection, the induction of luciferase activity was significantly less efficient (by nearly twofold) compared to the non-mutated reporter construct pGL4Luc-promLCT (Fig 2B). This observation endorsed the hypothesis of a functional role of the DR2 response element in the LCT gene promoter.

To further confirm the role and specificity of PPARγ in the control of LCT gene expression, we constructed a Caco-2 ShPPARγ cell line that stably expresses a short hairpin anti-sense RNA against PPARγ, leading to specific downregulation of PPARγ (Bouguen *et al*, 2015) (Appendix Fig S4). In these cells, both LCT gene transcription and activity were significantly reduced by 63 and 33%, respectively, compared to Caco-2 ShLuc control cells (Fig 2C). In addition, both GED and CLA-dependent induction of LCT activity were strongly compromised in PPARγ knockdown cells (Fig 2C). The induction of LCT expression by GED was also markedly reduced by GW9662, a specific PPARγ antagonist (Fig 2D). Moreover, LCT gene expression was significantly decreased in the proximal part of the small intestine of knockout mice presenting a specific deletion of PPARγ in IEC (PPARγ^ΔIEC KO mice; Fig 2E) compared to control animals, and PPARγ and LCT gene expression were significantly correlated in the duodenum and jejunum of wild-type Sprague Dawley rats (Fig EV4). Interestingly, we also observed that LCT and PPARγ proteins were co-expressed in the enterocytes of human duodenum (Appendix Fig S5). Finally, since some of the ligands used in our study are known to be able to weakly activate the PPARα receptor, we also assessed the potential involvement of this receptor in the control of LCT gene expression. We observed that fenofibrate, a specific PPARα agonist, was unable to modulate LCT activity and LCT gene transcription in Caco-2 cells (Fig 2F). Moreover, ChIP assays using a PPARα antibody did not result in the detection of fragment 8a–8b (Fig 2A and Appendix Fig S3). We also observed that PPARα expression was not modified in the ShPPARγ cell line or in small intestine of PPARγ^ΔIEC mice (Fig 2G) making participation of PPARα in the control of LCT expression in these two systems unlikely. Finally, in order to clarify definitively the possible involvement of PPARα in the modulation of LCT gene expression induced by our PPARγ agonists, we assessed LCT mRNA induction in Caco-2 cells treated with GW6471, a specific PPARα inhibitor (Fig 2H). No difference of LCT mRNA levels induction by GED, CLA, Pio, and Rosi was observed between GW6471-treated Caco-2 cells and control untreated cells (Fig 2H). Taken together, our results strongly suggest that PPARα activation is not involved in the control of LCT gene expression.

In mammals, the expression of the lactase gene is tightly regulated in a spatio-temporal manner. The lactase activity is usually greatest during the postnatal period and in infants, and then, lactase gene expression is downregulated after weaning. Although significant progress has been made, the underlying molecular mechanisms of this complicated pattern of expression are still incompletely understood. Several regulatory factors have been implicated in the control of lactase expression, such as Cdx-2 (caudal-related homeobox protein), nuclear receptors belonging to HNF (hepatocyte nuclear factor) family, GATA factors, or Oct-1 (Troelsen, 2005; Jarvela *et al*, 2009). Our data clearly demonstrate that PPARγ is also an essential factor controlling LCT gene expression. The overexpression of PPARγ in the duodenum and jejunum of not weaned rats compared to weaned animals (Fig EV4) suggests that PPARγ might be an important component of the molecular machinery involved in the maintenance of LCT expression before weaning. Moreover, a key finding of our study is also the first description of a pharmacological mechanism by which LCT gene expression is able to be modulated.

## Modulation of PPARγ increases LCT expression and activity *in vivo* in rodents and improves lactose intolerance symptoms in rats

To further explore *in vivo* the relationship between PPARγ and LCT, we assessed the potential induction of LCT gene expression by PPARγ modulators in rodents. Briefly, 30 mg/kg of GED or 200 mg/kg of CLA was administered daily by gavage for 7 days to weaned C57BL/6 mice or Sprague Dawley rats, and LCT activity and mRNA level were measured in the proximal part of the small intestine. Both PPARγ agonists significantly increased LCT expression and activity *in vivo* (Figs 3A and B, and EV5). This result led us to test whether modulating PPARγ was able to improve symptoms associated with lactose intolerance. For this purpose, weaned rats that are naturally LCT non-persistent (Fig EV4) were fed with a lactose-enriched diet (15 or 60% of total diet weight, Appendix Table S1). Compared to control animals, which received an isocaloric lactose-free diet, rats in the lactose groups lost weight, developed loose stools and diarrhea, and presented a significant increase in cecum weight and size (Appendix Fig S6, Fig 3C–E). Cecum dilatation reflected an increased fermentation activity of undigested lactose as revealed by the marked increase in total short-chain fatty acids (SCFA) concentration in the cecum contents of rats receiving lactose-enriched diet (Fig 3F). Stool consistency and cecum weight were rapidly and significantly improved by GED gavage in animals fed with a lactose-enriched diet (40 and 20% mean improvement, respectively, Fig 3D and E). GED treatment also significantly improved lactose-induced SCFA production in rat's cecum (Fig 3F).

This improvement in stool consistency along with the decrease in fermentation products and cecal dilatation obtained in GED-treated rats clearly suggests that modulating PPARγ activity might be clinically relevant to improve lactose maldigestion which usually associates diarrhea, abdominal pain, flatulence, and/or bloating after lactose ingestion in humans. Lactose intolerance prevalence cannot be evaluated with available data, but it has been estimated that hypolactasia, the LCT non-persistent phenotype, affects approximately 75% of the world's adult population with marked disparity according to ethnic origin (Sahi, 1994). Current management of lactose intolerance is based on exclusion of lactose intake. However, in addition to be naturally present in mammalian milk and dairy products, lactose is largely used in the food and pharmaceutical industry (as a bulking agent or vehicle), making nearly impossible the avoidance of this "hidden" lactose for patients. Moreover, the restriction of dairy products, often applied by individuals with lactose intolerance, fails to provide the daily recommended intake of calcium, leading to increased risk of osteoporosis, but also of development of obesity and cardiovascular diseases (Tremblay & Gilbert, 2011; Wang *et al*, 2012; Zhu & Prince, 2012). Our results suggest that it might be possible to restore lactose tolerance trough the modulation of PPARγ in hypolactasic patients and to overcome potential health concerns as well as the quality of life impairment associated with lactose intolerance.

A newly major objective in the field of nutrition research is the development of functional food products with demonstrated health benefits (Roberfroid, 2000). In this context, alternative to synthetic PPARγ agonists, that is, PPARγ modulators naturally present in diet or in medicinal plants, are currently a major topic of interest (Wang *et al*, 2014; Sauer, 2015). The ability of one of this natural PPARγ modulator, the trans-10, cis-12-conjugated linoleic acid (CLA)

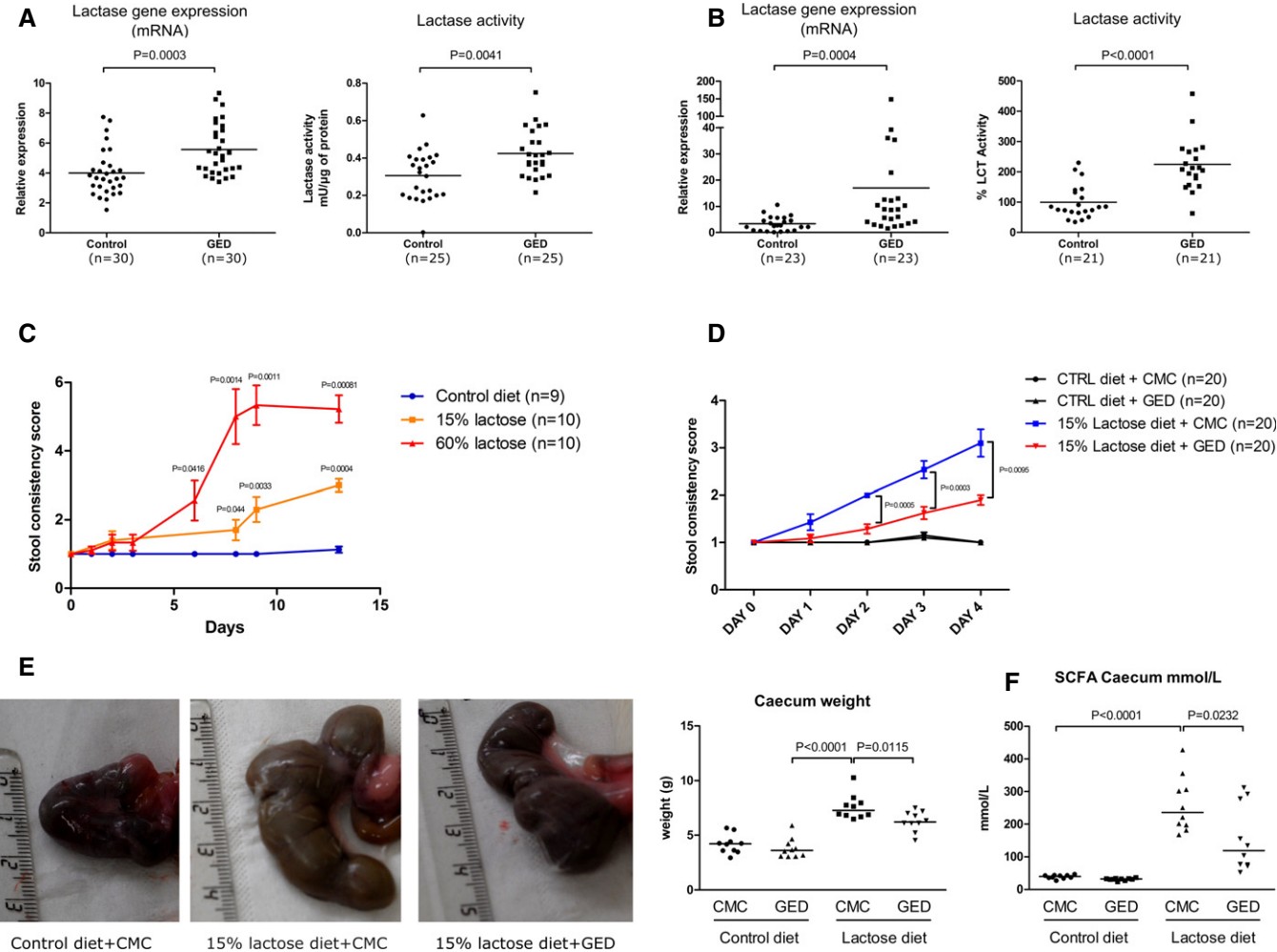

**Figure 3. LCT expression and activity *in vivo* in rodents following GED administration.**

A, B  LCT gene expression (qPCR) and LCT activity were assessed in the proximal small intestine of weaned C57BL/6 mice (A) and Sprague Dawley rats (B) treated with oral GED (30 mg/kg) for 7 days. Results represent the sum of three independent experiments. Horizontal bars represent mean values. LCT activities in rats are expressed as the percentage of activity compared to that measured in control animals (arbitrarily defined as 100%).

C  Stool consistency scores over time in weaned rats fed with lactose-enriched diets. *P*-values between lactose groups and control diet are indicated.

D  Stool consistency score in rats with and without lactose-enriched diet treated or not with GED. Results represent the sum of two independent experiments (*n* = 20 for each group).

E  Cecum dilatation induced by lactose diet was improved by GED treatment. Photographs show representative pictures of cecum size and morphology in the various groups. Results of cecum weight for one experiment (*n* = 10 for each group) are represented in the dot plot graph. Horizontal bars represent mean values.

F  Total SCFA concentration (mmol/l) in the cecal contents (*n* = 10 for each group). Horizontal bars represent mean values.

Data information: (C, D) Data are expressed as mean ± SEM. Statistical analysis: two-tailed nonparametric Mann–Whitney *U*-test.

isomer, to induce the LCT expression and activity (Figs 1A and D, and EV5) strongly suggests that PPARγ agonists naturally present in food sources might be promising for the management of lactose malabsorption.

Our results identified the gene encoding LCT as a new gene regulated by PPARγ and extend the pivotal role of PPARγ in the control of glucose homeostasis in the gut. We describe for the first time a pharmacological mechanism able to modulate LCT gene expression in LCT non-persistent phenotypic and genotypic contexts. We propose that controlling intestinal PPARγ activity by means of PPARγ ligands might improve lactose malabsorption and the symptoms of lactose intolerance.

# Materials and Methods

## Cell culture and treatment

Caco-2 cells were grown in Dulbecco's modified Eagle's medium (DMEM, Invitrogen, Life Technologies, Cergy-Pontoise, France) supplemented with 10% fetal calf serum (FCS, Dutscher, Brumath, France), 1% penicillin-streptomycin (5 ml/l) (Invitrogen, Life technologies), and 1% non-essential amino acids (5 ml/l) (Invitrogen, Life technologies). All cell lines were cultured as confluent monolayers at 37°C in a controlled, 5% $CO_2$ atmosphere. Caco-2 cells spontaneously differentiate

to a small bowel phenotype after confluency (Ding *et al*, 1999).

For cell stimulations, $1 \times 10^6$ cells per well were seeded in six-well plates. Serum deprivation was used 16 h prior to stimulation in order to synchronize the cells. Cells were treated with various concentrations of GED (Nogra Pharma Ltd, Italy), pioglitazone (Sigma-Aldrich), 5-ASA (Sigma-Aldrich), CLA (Sigma-Aldrich), rosiglitazone (Sigma-Aldrich), or fenofibrate (Sigma-Aldrich). The PPARγ antagonist GW9662 (10 μM) and PPARα antagonist GW6471 (10 μM) were used concomitantly with the agonist. When necessary, the DMSO vehicle (Sigma-Aldrich) was used as control. After 24 h of stimulation, cells were washed three times with sterile PBS before RNA extraction. Cell stimulations were performed in four replicates for microarray analysis and in three, four, or six replicates for other stimulations.

## RNA extraction

Total RNA was extracted with a Nucleospin RNA kit (Macherey-Nagel, Hoerdt, France) according to manufacturer's instructions. After RNAse inactivation, total RNA was cleaned of genomic DNA traces by DNAse treatment and eluted in RNAse-free DEPC-treated water. The purity of the RNA was evaluated by UV spectroscopy on a Nanodrop system (Nyxor Biotech, Paris, France) from 220 to 350 nm. Before microarray experiments, RNAs were also profiled on an Agilent 2100 bioanalyzer. One μg of total RNA with a minimum concentration of 50 ng/μl was used in the microarray and qRT–PCR analysis.

## Microarrays

Dual-color gene expression microarrays were used to compare the cRNA from the samples. 44,000 genes were screened. The RNAs from the samples were first reverse-transcribed into cDNA (Affinity-Script RT, Agilent), which were then used as the substrate for the synthesis and amplification of cRNA by T7 RNA polymerase in the presence of cyanine 3-CTP for the CTL sample (green fluorescence) and cyanine 5-CTP for the PPARγ agonist sample (red fluorescence). The two-labeled cRNAs were mixed, hybridized on the same array (G4851A Agilent 8 × 44K), and then scanned (with an Agilent G2505B scanner). Fluorescence was visualized after laser excitation and the relative intensities of the two fluorophores were expressed as a ratio, in order to yield the over- or under-expression status of each gene (using GeneSpring software (Agilent)). This analysis was performed for each PPARγ agonist. Data have been deposited in NCBI's Gene Expression Omnibus (Edgar *et al*, 2002) and are accessible via GEO Series accession number GSE68852 (http://www.ncbi.nlm.nih.gov/geo/query/acc.cgi?acc = GSE68852).

## Quantitative PCR

Expression of genes of interest was quantified by quantitative PCR of corresponding reverse-transcribed mRNA. One μg of total RNA was reverse-transcribed into cDNA using the High Capacity cDNA Archive kit (Applied Biosystems). Amplification was performed using an ABI PRISM 7000 sequence detection system (Applied Biosystem) using Power SYBR® Green PCR master Mix (Applied Biosystem). Primer pairs for each human transcript were chosen

with qPrimer depot software (http://primerdepot.nci.nih.gov). See Appendix Table S2 for the oligonucleotides used in this study. Quantification of qPCR signals was performed using ΔCt relative quantification method using GAPDH as a reference gene for human and rat samples and β-actin for mouse samples. Values were represented in terms of relative quantity of mRNA level variation or fold increase compared to control conditions.

## Western blot analysis

$6 \times 10^6$ Caco-2 cells were treated 24 h with PPARγ agonists, 5-ASA (30 mM), Pio (1 μM), Rosi (1 μM), GED (30 mM), and CLA (1 mM), and then, total protein was extracted in RIPA buffer containing 50 mM Tris–HCl pH 7.6, 150 mM NaCl, 1.5 mM MgCl$_2$, 5 mM EDTA, 1% Triton X-100, and 10% glycerol, supplemented with 100 mM sodium fluoride (NaF), 2 mM sodium orthovanadate (Na$_3$VO$_4$), 10 mM sodium pyrophosphate (NaPPi), 1 mM phenyl-methanesulfonyl fluoride (PMSF), and a classical protease-inhibitor cocktail (Complete Mini, EDTA-free, Roche). 50 μg of protein for each sample was separated by SDS–PAGE electrophoresis and transferred in 100% pure nitrocellulose membranes. After two washes in TBS-Tween buffer containing 30 mM Tris–HCl pH 8, 300 mM NaCl, and 0.1% Tween-20, membranes were blocked 2 h in 5% non-fat dry milk in TBS supplemented by 0.05% Tween-20 and probed 2 h a RT with primary antibodies directed against lactase (Novus Biological, monoclonal mouse, clone 3C105.1, 1:200), PPARγ (Cell Signaling, monoclonal rabbit, clone C$_6$H$_{12}$, 1:1,000), and β-actin (Sigma, monoclonal mouse, clone AC-15, 1:5,000) diluted in 1% non-fat dry milk in 0.05% TBS-Tween. Membranes were then incubated with secondary horseradish peroxidase-conjugated antibodies [anti-rabbit (Jackson ImmunoResearch) and anti-mouse (Sigma)], 1:10,000 in 1% non-fat dry milk in 0.05% TBS-T for 1 h at room temperature.

Immunodetection was performed with SuperSignal West Pico chemiluminescent substrate (Thermo Scientific Pierce, Erembodegem) according to manufacturer's protocol. Chemiluminescent signals were captured by a cooled charged coupled device (CCD) camera, and the optical density of target bands was determined using a computer-assisted densitometer and the ImageJ public domain software (W. S., Rasband, ImageJ, U. S. National Institutes of Health, Bethesda, MD; http://rsb.info.nih.gov/ij/, 1997–2011). Protein levels for each sample were expressed as units of optical density (OD) per quantity of total proteins and normalized with the β-actin, and results were expressed as fold change compared to the control groups.

## Immunofluorescence and immunohistochemistry

### *Immunofluorescence*
After 24 h of stimulation by 1 mM GED, 1 μM Pio, 1 μM Rosi, and/or 1 mM CLA, Caco-2 cells were fixed in 4% paraformaldehyde for 10 min, permeabilized in 0.1% Triton X-100 phosphate-buffered saline (PBS) for 15 min at room temperature, and then incubated with blocking buffer [3% bovine serum albumin (BSA) in PBS]. Incubation with primary lactase antibody (HPA007408 from Atlas Antibodies; dilution 1:200) was performed overnight 4°C. Normal Goat IgG (Invitrogen) was used as negative control. Incubation with secondary antibody (goat anti-rabbit Alexa 488 labeled (Invitrogen); dilution 1:100) was performed 1 h at room temperature. Nuclei

were stained with Hoechst 33342 solution (0.125 mg/ml) (Sigma-Aldrich). Cells were visualized under a fluorescence microscope (Leica, Bensheim, Germany).

Histological sections of human duodenal biopsies (our biological collection of human intestinal specimens: a local ethics committee approved the study and all subjects gave informed consent (No. DC-2008-642); the experiments conformed to the principles set out in the WMA Declaration of Helsinki and the Department of Health and Human Services Belmont Report) were examined for PPARγ and lactase staining. Tissue specimens were fixed in fresh 4% paraformaldehyde (PFA)/PBS solution for 3 h at room temperature, were dehydrated in a graded ethanol series, and embedded in low-temperature-fusion paraffin. 4-μm-thick sections were treated 10 min with an endogenous peroxidase blocking kit (GeneTex) and 15 min with a Streptavidin/Biotin Blocking Kit (Vector Labs). Then, sections were restored by 10 min of incubation in TBS supplemented with 0.1% Triton X-100, at 4°C and blocked 10 min at RT with 5% calf serum in TBS supplemented by 0.05% Tween-20 and 10 min at RT with 3% BSA in 5% non-fat dry milk. Sections were incubated overnight at 4°C with monoclonal mouse primary anti-lactase or polyclonal rabbit anti-PPARγ (both purchased from Novus Biological) at the dilution of 1:100 and 1:50, respectively. After two washes in 0.05% Tween in TBS, sections were incubated 1 h at RT with biotinylated secondary antibody diluted 1:1,000 (donkey anti-mouse purchased from Jackson Immuno Research) in 0.05% Tween in TBS and 30 min at RT with Streptavidin-HRP diluted 1:2,000. The specific proteins were detected as brown precipitates obtained by a short incubation (3–5 min) with 3,3′-Diaminobenzidine (DAB) (Dako LSAB Corporation), a chromogen substrate for peroxidase enzyme. Sections were counterstained with Harris hematoxylin and observed under the Leica DM2000 light microscope at 40× magnification.

**Lactase activity**

Lactase activity was evaluated by using a glucose oxidase method (Glucose Assay Kit, Sigma) previously described by Dahlqvist (1968). This lactase assay is based on the measurement of the amount of glucose produced following the action of lactase by incubating samples with a lactose buffer solution (0.056 mol/l lactose in a 0.1 mol/l Na-maleate buffer). For Caco-2 cells, lactase activity was determined directly from the cell monolayer. After extensive washing, the cell monolayer was incubated with lactose buffer for 1 h at 37°C. The supernatant was recovered; 50 μl was diluted with 100 μl of glucose oxidase reagent and incubated at 37°C for 1 h. The reaction was stopped with 100 μl of $H_2SO_4$ and read by spectrophotometry at 450 nm. When lactase activity was determined from intestinal sample, tissue samples were first dounce-homogenized in 0.9% NaCl on crushed ice. These homogenates were then diluted in 0.9% NaCl (1/500), and 50 μl of dilution was incubated with lactose buffer and used to determine lactase activity. For each experiment, the background attributed to the remaining glucose in the samples was measured by incubating cells or cell extracts in lactose-free buffer.

**Glucose uptake assay**

Glucose uptake was evaluated by using the glucose uptake colorimetric assay kit (Sigma-Aldrich) according to the manufacturer's

instructions. Briefly, Caco-2 cells were seeded into a 96-well plate at a density of 30,000 cells per well. Serum deprivation was used 16 h prior to stimulation in order to synchronize the cells. Cells were treated with GED (1 mM) or pioglitazone (1 μM) for 24 h. Cells were then washed three times with PBS and were glucose-starved by incubating with 100 μl of KRPH buffer (Krebs-Ringer-Phosphate-HEPES (KRPH) buffer—20 mM HEPES, 5 mM $KH_2PO_4$, 1 mM $MgSO_4$, 1 mM $CaCl_2$, 136 mM NaCl, and 4.7 mM KCl, pH 7.4) containing 2% BSA for 40 min. 10 μl of 2-deoxyglucose (2-DG; 10 mM) was then added, and incubation was continued for 20 min. 2-DG is taken up by the cells and phosphorylated by hexokinase to 2-DG6P, which cannot be further metabolized and accumulates in cells. Following incubation, cells were washed three times with PBS and lysed with 80 μl of the extraction buffer provided. The amount of 2-DG6P (which is directly proportional to glucose uptake by the cells) was determined by a colorimetric detection assay according to the manufacturer's protocol.

**Generation of PPARγ knockdown cells**

Generation of PPARγ knockdown Caco-2 cells was described in Bouguen et al (2015).

**Reporter gene assay**

The fragment corresponding to the first 321 bp upstream to the transcription initiation site of the human lactase gene was obtained by PCR from genomic DNA of Caco-2 cells using "Hs-Prom-0.3Kb sens" and "Hs-Prom-0.3Kb anti-sens" oligonucleotides. The PCR products were cloned into a TOPO pCR4 vector (TOPO TA cloning, Invitrogen) and then sequenced in order to check for potential Taq Polymerase errors. A mutation-free fragment was subcloned into the vector pGL4.10 [Luc2] (Promega) using the *XhoI/HindIII* restriction sites introduced into the oligonucleotides. This construction and the empty vector control were transiently transfected in Caco-2 cells using Nucleofector™ Technology (Solution SE, program DS 123). Transfected cells were treated with PPARγ agonist for 24 h. Luciferase activity was measured using the luciferase assay kit (Promega) in a Wallac Victor2™ 1420 multilabel counter (Perkin Elmer).

**Site-directed mutagenesis**

Reporter constructs "pGL4LucPromLCT MUT" and "pGL4LucPromLCT DEL" were generated using the QuickChange® Site-directed Mutagenesis Kit (Stratagene) according to the manufacturer's instructions, using "pGL4LucPromLCT" vector as template and "0.3 kb Mut Fwd"/"0.3 kb Mut Rev" oligonucleotides for "pGL4LucPromLCT MUT" synthesis and "0.3 kb Del Fwd"/"0.3 kb Del Rev" oligonucleotides for "pGL4LucPromLCT DEL" synthesis. Following sequence verification, positive clones were used directly in transfection assays.

**Chromatin immunoprecipitation experiments**

The physical binding of PPARγ onto the LCT gene promoter was studied by ChIP experiments in Caco-2 cells ($5 \times 10^6$ cells) stimulated for 24 h with 1 mM GED in 100 mm cell culture petri dishes.

Caco-2 cells were synchronized by the addition of serum-free medium for 16 h and then stimulated for 24 h using the protocol described previously. Cells were then rinsed with PBS, and the protein–DNA complex was fixed by adding 1% PFA for 30 min at room temperature. This binding was stopped by the addition of glycine (0.125 M). Cells were collected by scrapping in cold PBS and protease inhibitors (Sigma). The cell pellet obtained by centrifugation was taken up in 300 μl SDS buffer (1% SDS, 10 mM EDTA, 50 mM Tris–HCl pH 8, protease inhibitors) and sonicated (Diagenode, Bioruptor UCD-200 TM-EX) for 30 s, followed by 30-s resting time, during 10 min. For each immuno-precipitation, 125 μl of cross-linked sonicated sample was diluted with 225 μl of IP buffer (1% Triton X-100, 150 mM NaCl, 2 mM EDTA, 20 mM Tris–HCl pH 8.1, and protease inhibitors) and precleared for 4 h by adding 40 μl of protein A/G beads (50% slurry protein A/G Sepharose, Clinisciences) and 5 μg of salmon sperm DNA (Invitrogen). Complexes were immunoprecipitated with 2 μg of specific antibodies (PPARγ, mouse monoclonal IgG2a, clone K8713, R&D Systems; PPARα, mouse monoclonal antibody, clone 3B6, Thermo Fisher Scientific) by incubation overnight at 4°C under rotation. Immune complexes were recovered by adding 40 μl of protein A/G Sepharose (50%) plus 2 μg salmon sperm DNA and incubated for 4 h at 4°C. The beads were washed twice in wah buffer 1 (0.1% SDS, 1% Triton X-100, 150 mM NaCl, 0.1% Deoxycholate, 1 mM EGTA, 2 mM EDTA, 20 mM Tris–HCl pH 8.0), twice in wash buffer 2 (0.1% SDS, 1% Triton X-100, 500 mM NaCl, 0.1% Deoxycholate, 1 mM EGTA, 2 mM EDTA, 20 mM Tris–HCl pH 8.0), once in wash buffer 3 (0.25 mM LiCl, 0.5% Deoxycholate, 0.5% NP-40, 0.5 mM EGTA, 1 mM EDTA, 10 mM Tris–HCl pH 8.0), and three times in wash buffer 4 (1 mM EDTA, 10 mM Tris–HCl pH 8.0). The co-immunoprecipitated DNA was then extracted with 150 μl of extraction buffer (0.1 M NaHCO3, 1% SDS). Cross-linking was reverse overnight at 65°C. DNA was then purified using the PCR Clean-up kit (Macherey-Nagel) and analyzed by PCR.

### Animal experimentation

Animal experiments were performed in the accredited Pasteur Institute animal care facility (Institut Pasteur de Lille, France; no B59-35009) according to governmental guidelines (no 2010/63/UE; Décret 2013-118) and animal ethics committee approval (protocol no 05273.01). Specific pathogen-free male C57BL/6 mice and Sprague Dawley rats were obtained from Janvier Labs (France). Mice and rats were housed five animals/cage and three animals/cage, respectively, in a specific pathogen-free facility, in an air-conditioned room with controlled temperature ($22 \pm 1$°C), humidity (65–70%), and 12-h light/12-h dark cycles. Animals were fed with standard laboratory chow (except when indicated) and were provided with autoclaved tap water *ad libitum*. Animals were acclimatized for at least 1 week before entering the study.

In order to assess the effect of GED and CLA on lactase expression and activity, weaned C57BL/6 mice (8 weeks old) and weaned Sprague Dawley rats (older than 2 months) were randomized into two groups receiving daily intragastric gavage of 30 mg/kg of GED, 200 mg/kg of CLA, or vehicle (0.5% CMC, 1% Tween-80). After 7 days of treatment, animals were euthanized and the gastrointestinal tract was removed via a midline laparotomy. Approximately

0.5 cm of proximal intestine tissue specimens was snap-frozen for further extractions. LCT mRNA expression and LCT activity were assessed as described above.

The effect of GED on the symptoms associated with lactose intolerance was evaluated in weaned rats fed with a lactose-enriched diet provided by Ssniff Spezialdiäten GmbH (Soest, Germany; Appendix Table S1). Animals were monitored daily, weighed and stool consistencies were evaluated.

C57BL/6 mice carrying a targeted disruption of the gene encoding PPARγ in IECs were generated by breeding mice harboring a floxed PPARγ (PPARγ$^{fl/fl}$) (Imai *et al*, 2004) to transgenic mice bearing a tamoxifen-dependent Cre recombinase (vil-Cre-ERT2) expressed under the control of the villin promoter (El Marjou *et al*, 2004). Recombination and PPARγ gene deletion were induced by tamoxifen treatment. These mice were designated "PPARγ$^{ΔIEC}$ KO mice". Mice received an intraperitoneal injection of tamoxifen (10 mg/ml; 100 μl) for 5 consecutive days and sacrificed. Control animals correspond to littermate control mice which do not carry transgenic Cre recombinase but received injection of tamoxifen. These mice were designated "Control mice". Intestinal samples were collected ("PPARγ$^{ΔIEC}$ KO mice": $n = 4$, three females and one male; "Control mice": $n = 4$, three females and one male) and snap-frozen for further extraction.

### SCFA quantification

Short-chain fatty acids were extracted and measured as described in Alexandre *et al* (2013).

### Genotyping

Lactase genotyping of C/T$_{13910}$ and G/A$_{22018}$ polymorphisms for Caco-2 cells were determined as described in Matthews *et al* (2005).

### Statistics

The data are presented as mean with SEM or SD. All graphs were plotted and analyzed with GraphPad Prism 5 Software (San Diego, CA, USA) using a two-tailed nonparametric Mann–Whitney test. Statistical tests were validated with the support of a statistician. *P*-values < 0.05 were considered statistically significant, and exact *P*-values were indicated (except when $P < 0.0001$).

### Data availability

Microarray data are available at NCBI's Gene Expression Omnibus (GEO Series accession number GSE68852; http://www.ncbi.nlm.nih.gov/geo/query/acc.cgi?acc = GSE68852.

**Expanded View** for this article is available online.

### Acknowledgements

We thank Céline Villenet, Sabine Quief, and Frédéric Lepretre (Functional and Structural Genomic Platform, Université de Lille) for conducting microarray experiments and the management of microarray dataset. We acknowledge the support of the DigestScience Foundation. This study was supported by Institut National de la Santé et de la Recherche (Inserm), *Société Nationale Française de Gastro-Entérologie* (FARE 2015) and NograPharma Ltd, Dublin.

**The paper explained**

**Problem**

Lactose intolerance is a frequent condition that causes abdominal discomfort, pain, and diarrhea. It results from lactase (LCT) enzyme deficiency (hypolactasia) produced by IECs. It is estimated that hypolactasia affects approximately 75% of the world's adult population. Except for lactose-free diet, no treatment can cure lactose intolerance and the regulation of LCT enzyme expression remains poorly understood. PPARγ is a nuclear receptor expressed by IECs playing a key role in gut homeostasis and metabolism regulation.

**Results**

We identified LCT gene as one of the master regulated gene by PPARγ modulators in IECs. We demonstrated that both synthetic and natural PPARγ agonist ligands are able to increase the expression and activity of LCT *in vitro* and *in vivo*. We also developed a new model of lactose intolerance in weaned rodents and demonstrated that PPARγ agonist improved symptoms induced by lactose-enriched diet.

**Impact**

We identified for the first time a pharmacological mechanism able to modulate LCT expression and activity. We propose that modulating intestinal PPARγ activity by means of PPARγ ligands might restore lactose tolerance and might improve lactose malabsorption and the symptoms associated with hypolactasia.

## Author contributions

MFu, SS, A-MD, CR, LD, PD, and BB designed the study. MFu, SS, AL, A-MD, CD, MG, AMM, and BB performed experiments and analysis of the data. MFi performed and analyzed microarray experiments. DM managed and provided transgenic mice. MFu, SS, A-MD, DM, LD, J-FC, PD, and BB wrote the study and participated to the critical reading of the manuscript.

## Conflict of interest

The authors declare that they have no conflict of interest.

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
