## [Review Process File · EMBO Molecular Medicine]

Peroxisome proliferator-activated receptor gamma (PPAR γ) regulates lactase expression and activity in the gut

Mathurin Fumery, Silvia Speca, Audrey Langlois, Anne-Marie Davila, Caroline Dubuquoy, Marta Grauso, Anthony Martin Mena, Martin Figeac, Daniel Metzger, Christel Rousseaux, Jean-Frédéric Colombel, Laurent Dubuquoy, Pierre Desreumaux, Benjamin Bertin

Corresponding author: Benjamin Bertin, Université de Lille - Inserm

Review timeline:

Submission date:	13 March 2017
Editorial Decision:	24 April 2017
Revision received:	22 July 2017
Editorial Decision:	24 August 2017
Revision received:	29 August 2017
Accepted:	01 September 2017

Transaction Report:

Editor: Céline Carret

1st Editorial Decision

24 April 2017

Thank you for the submission of your manuscript to EMBO Molecular Medicine. We have now heard back from the two referees whom we asked to evaluate your manuscript.

You will see in the reports below that both referees find the topic of interest. However they both have suggestions to increase the conclusiveness and mechanistic insights of the study that we would like you to follow, experimentally when necessary, in your revised article.

Please note that EMBO Molecular Medicine encourages a single round of revision and that, as acceptance or rejection of the manuscript will depend on another round of review, your responses should be as complete as possible.

Revised manuscripts should be submitted within three months of a request for revision; they will otherwise be treated as new submissions, except under exceptional circumstances in which a short extension is obtained from the editor.

I look forward to receiving your revised manuscript.

***** Reviewer's comments *****

Referee #1 (Remarks):

In the present study, the authors aimed to characterize PPAR γ target genes involved in intestinal metabolic functions. By microarray analysis, the authors identified the LCT gene as the most upregulated gene by PPAR γ agonists in Caco-2 cells. Although the role of PPAR γ in lactose metabolism is potentially interesting, it is still unclear whether PPAR γ directly binds and regulates LCT gene expression. In addition, some of the data quality is poor.

Major comments

1. Usage of uncharacterized PPAR γ agonists

Throughout the manuscript, the authors used several agonists that are not well-characterized. They should therefore test whether Rosiglitazone (0.5 μ M), a widely used PPAR γ ligand, can induce LCT gene expression and activity.

2. Data quality issues

(1) In figure 1B, the immunofluorescence images of LCT are not clear. Is it localized to the nucleus or the cytoplasm? The authors should provide pictures with better resolution.

(2) In figure 1C, the authors should indicate the molecular weight and include the PPAR γ blot.

(3) In figure 2, the quality of the ChIP assay is poor. The authors should either perform PPAR γ ChIP-Seq in Caco-2 cells or at least use a published data set to determine whether PPAR γ indeed directly binds to the LCT gene locus.

(4) The error bars in Figure 2B and 2E are huge.

3. Better method for in vivo experiment

In figure 3, the authors used a stool consistency score, which is a very subjective measurement. Can they use a more objective measurement, for example, water content in stool?

Referee #2 (Remarks):

This manuscript presents a novel discovery that directly connects the nuclear receptor PPAR γ with Lactose gene in the enterocytes. The authors did a nice job in the molecular biology and biochemistry parts of the story. Nevertheless, few experiments are needed at this stage.

1. Study on colocalization of PPAR γ and LCT in the crypt-to-villus axis of the intestinal mucosa is necessary. Is PPAR γ expressed in the differentiated enterocytes that express LCT?

2. The authors identified the DR2 at 8a-8b position of the LCT promoter as the functional one. They should definitively present data with the full length 3Kb promoter with the 8a-8b mutated DR2 and show that this is the only one that is working (i.e. there are other DR2 and DR1 upstream).

3. The authors are invited to compare the human with the rat promoter of LCT. Indeed, they present human and rat data but they do not show which response element is conserved on the promoter.

4. The ChIP experiments deserve more control data, for example using PPAR α -PPAR β antibodies in the immunoprecipitation as negative controls.

5. The manuscript is lacking the translational part. Would diabetic patients take use PPAR γ agonists benefits for lactose intolerance?

6. The authors used shPPAR γ and in the supplementary shPPAR α . They are invited to include in the main manuscript an experiment in cells with shPPAR α , shPPAR γ , shPPAR β with and without

specific agonists for the 3 receptors and detect modulation of LCT mRNA levels. This is necessary to prove the specificity of gamma receptor.

1st Revision - authors' response

22 July 2017

Referee #1

Major comments

1. Usage of uncharacterized PPARgamma agonists

Throughout the manuscript, the authors used several agonists that are not well-characterized. They should therefore test whether Rosiglitazone (0.5uM), a widely used PPARgamma ligand, can induce LCT gene expression and activity.

Thank you for the suggestion. We agree with this comment and the effect of rosiglitazone was assessed in several independent experiments (3 independent experiments, $12 < n < 15$). Rosiglitazone ($1 \mu\text{M}$) was able to significantly increase LCT mRNA expression (12 fold) and activity (3.5 fold) in Caco-2 cells, which correspond approximately to the effect observed with pioglitazone. We also observed that rosiglitazone increased LCT protein expression (in immunofluorescence and western blot assays). All these results are now included in Figure 1. We also done a dose-effect of rosiglitazone on LCT mRNA expression in Caco-2 cells and added the results to what is now referred as "Figure expanded view 1" (Fig EV1) (previously Supplemental Figure S1). In the revised version of our manuscript, Fig EV1 gathers all the data concerning dose-effect of GED, CLA, pioglitazone and rosiglitazone on LCT expression in Caco-2 cells. We finally checked the effect of rosiglitazone on other disaccharidases in Caco-2 cells and, as for pioglitazone and GED, found that rosiglitazone did not induce sucrose-isomaltase nor maltase-glucoamylase mRNA expression. These results were added to what is now referred as "Figure expanded view 2" (Fig EV2) (previously Supplemental Figure S3). The "Results" and "Material and methods" sections were both modified in the revised version of the article to include all these new data.

2. Data quality issues

(1) In figure 1B, the immunofluorescence images of LCT are not clear. Is it localized to the nucleus or the cytoplasm? The authors should provide pictures with better resolution.

All our apologies for the poor quality of pictures in figure 1B. We built a new panel with pictures of higher resolution. We added the results of immunofluorescence obtained with rosiglitazone to this new panel. Moreover, we are also now providing pictures of lactase immunostaining in Caco-2 of magnification X40 for each condition as "SourceDataForFigure 1B" files in high quality pdf. These pictures clearly show that, as expected, LCT is not localized in the nucleus of Caco-2 cells, but is rather expressed in the cytoplasm and/or cytoplasmic membrane. If necessary, we can of course supply the original files in high resolution jpg format to the reviewers and/or to the readers as "SourceDataForFigure 1B".

(2) In figure 1C, the authors should indicate the molecular weight and include the PPARgamma blot.

We agree with this comment. We chose to complete and improve LCT protein expression analysis by testing LCT protein induction by PPARg agonists in Caco-2 cells using western blot. We found that all tested PPARg agonists were able to increase LCT protein expression. This result of western blot analysis is now replacing LCT immunoprecipitation assay in Figure 1C. We also provide unedited pictures of full scanned membrane as "SourceDataForFigure 1C" file. The "Results" and "Material and methods" sections were both modified in the revised version of the article to include this new result. Nevertheless, we also reproduced the immunoprecipitation assays with protein extracts from Caco-2 cells stimulated with 5-ASA, GED, Pio, Rosi and CLA. Similar results were observed with this approach. Thus, we have chosen to show western-blot results in the revised version of the manuscript, but of course, we will consider all suggestions concerning the possibility to show IP results in the new manuscript. We think that this new data improve our message and confirm that PPARg modulators induce LCT expression. We also hope that this properly answer the comment of the reviewer.

(3) In figure 2, the quality of the ChIP assay is poor. The authors should either perform PPARgamma ChIP-Seq in Caco-2 cells or at least use a published data set to determine whether PPARgamma indeed directly binds to the LCT gene locus.

Thank you for this suggestion. We are fully agree that PPARg ChIP-Seq in Caco-2 cells would be a very interesting approach to better understand the role of PPARg in intestinal epithelial cells (at least in Caco-2 cells). Unfortunately, the 3-month period required for paper reviewing did not allow us to develop this approach in terms of time and availability of technical means and analysis. But, again, thank you for this suggestion and we will try to develop these experiences as soon as possible. As recommended by the reviewer, we also looked for published data set of PPARg ChIP-seq analysis in Caco-2 cells or in other intestinal epithelial cells lines, but we did not find any published data.

Concerning the quality of our ChIP assay, and in relation with the comments and suggestions of the reviewer n°2, we improved the data by including new controls (IP control with beads alone, IP control with PPARa antibody). We included these new data in figure 2A. We also added some results of ChIP assay obtained with Caco-2 cells stimulated with pioglitazone (shown as “Appendix Figure S3” in the revised version of the manuscript). Please, see reply to reviewer#2’s comment n°2 for more details.

(4) The error bars in Figure 2B and 2E are huge.

New experiments and analysis were done and significantly improved the data. Figure 2B and 2E were modified according to these new results.

3. Better method for in vivo experiment

In figure 3, the authors used a stool consistency score, which is a very subjective measurement. Can they use a more objective measurement, for example, water content in stool?

We understand this comment. Unfortunately, we do not have this kind of data as we did not evaluate stool water content. Macroscopic evaluation of stool form and consistency is widely used in both clinical and experimental field, notably with the Bristol Stool Form Scale. As a translational research lab, involving numerous gastroenterologists, we have established our stool scoring system based on the Bristol Stool Scale with the help of gastroenterologists from the lab. We can assure you of the greatest rigor with which stool consistency was scored, in a blind manner by two very experienced investigators. Moreover, the symptoms induced by lactose-enriched diet and the effects of GED in rats fed with lactose-enriched diet were also assessed by objective measurements such as caecum weight and SCFA concentration in the caecum.

Referee #2 (Remarks):

This manuscript presents a novel discovery that directly connects the nuclear receptor PPARgamma with Lactose gene in the enterocytes. The authors did a nice job in the molecular biology and biochemistry parts of the story. Nevertheless, few experiments are needed at this stage.

We sincerely thank the reviewer for his/her positive feedback.

1. Study on colocalization of PPARgamma and LCT in the crypt-to-villus axis of the intestinal mucosa is necessary. Is PPARgamma expressed in the differentiated enterocytes that express LCT?

This is indeed a very interesting point, thank you for this comment. We thus conducted immunohistochemistry experiments in order to detect LCT and PPARg proteins in the differentiated enterocytes of the human duodenum. We used paraffin sections from human duodenal biopsies (n=3). We did not co-localize LCT and PPARg proteins in the same section because we used DAB substrate for both antibodies staining, but we detected each protein in consecutive, adjacent sections of the same biopsy. The results are now included in the revised version of the manuscript and we

propose to show them in “Appendix Figure S5” as we observed a nice co-expression of LCT and PPARg proteins in the enterocytes of the duodenum villi.

2. The authors identified the DR2 at 8a-8b position of the LCT promoter as the functional one. They should definitively present data with the full length 3Kb promoter with the 8a-8b mutated DR2 and show that this is the only one that is working (i.e. there are other DR2 and DR1 upstream).

In relation with Reviewer#1’s comment n°3 and Reviewer#2’s comments n°2 and n°4, we modified and improved ChIP and transfection data as follow:

- ChIP experiment was redone with GED and Pio, and negative controls were added (IP control with beads alone, IP control with PPARa antibody). In the revised version of the manuscript, the results obtained with GED are included in Figure 2A and we propose to include the results obtained with pioglitazone as “Appendix Figure S3”.
- The potential binding of PPARg on other putative identified response elements were retested. Only response elements predicted at least by two different algorithms were retained for this analysis (we added a sentence in the legend of Appendix figure S2; compared to our previous analysis, most of the PPRE were predicted by at least two different algorithms and only one DR1 and one DR2 were not selected according to this criterion). As you can see on the figure below, no other significant binding of PPARg on LCT promoter was identified by ChIP.

- As a starting point, the choice of the 3kb-length sequence to be analyzed was quite arbitrary. Because there was no evidence for binding to other potential PPRE (other than the identified DR2 8a-8b), we conducted the suggested mutagenesis approach with our “pGL4Luc-promLCT construct” of 321bp-length in order to determine whether this DR2 was functional. The results were included in figure 2B. Two different constructions were created by site-directed mutagenesis (i) one with three mutated nucleotides and (ii) one with three deleted nucleotides (see figure 2B). Although these two mutated constructs were still responsive to GED in transient transfection, the induction of luciferase activity was significantly ($P=0.002$) less efficient compared to the non-mutated reporter construct pGL4Luc-promLCT, suggesting that the DR2 response element is functional in the LCT gene promoter.

It is well known that the 150 bp region of the lactase promoter upstream of the transcriptional start site is important for the regulation of lactase expression (this region is notably the only well conserved sequence between rat, mouse and human; see below response to comment n°3). This promoter region binds transcription factors (Cdx2, HNF1, GATA factors) that are very important for lactase expression but also for the differentiation of the intestinal epithelial cells. Therefore, one hypothesis to explain the remaining GED-dependent activity of our mutated constructs might be that

PPAR γ activation could sustain/increase the expression/activity of transcription factors involved in the regulation of this 150 bp promoter region.

3. The authors are invited to compare the human with the rat promoter of LCT. Indeed, they present human and rat data but they do not show which response element is conserved on the promoter.

As mentioned above, it is well known that the only conserved region of the lactase promoter between human, mouse, rat, rabbit and pig is a proximal 150 bp sequence just upstream of the transcriptional initiation site (Troelsen *JT BBA* 2005; 1723: 19-32). Our own alignment strategy using Martinez/Needleman–Wunsch algorithm (MegAlign module of DNASTAR Lasergene software; DNASTAR Inc., Madison, WI, USA) and ConTra alignment tool (Nucleic Acids Res. 2011 Jul;39(Web Server issue):W74-8) confirmed these observations (we can provide pictures if necessary). This 150 bp region does not contain any PPRE.

The transcriptional regulation of the lactase gene is not based solely on this 150 bp proximal region, but it is also recognized that more distal regulatory sequences are involved (Troelsen *JT BBA* 2005; 1723: 19-32; Lee *SY et al. J. Biol. Chem.* 277 (2002)13099– 13105 for example). Despite a similar LCT expression pattern in mammals (eg inhibition of LCT expression after weaning) and a certain degree of conservation in the mapping of these regulatory regions, there is no sequence identity between the human, mouse, rat and pig genes within the distal LCT promoter sequences. This therefore suggests that the apparent lack of conservation of PPRE sequence in the promoters of human and rat genes does not necessarily mean that the PPAR γ -dependent regulatory mechanism cannot be conserved between the two species.

So, we understand the reviewer's comment and we agree that it probably constitutes an interesting question, but we will not be able to detail the mechanisms of PPAR γ -induced expression of the LCT gene in rats or mice (as it would require time to characterize them). Moreover, as explained above, our feeling is that this lack of conservation between species does not question the validity of our data obtained in mice and rats treated with PPAR γ agonists.

4. The ChIP experiments deserve more control data, for example using PPAR α -PPAR β antibodies in the immunoprecipitation as negative controls.

As explained above (reply to comment n°2), PPAR α antibody was used as negative control in ChIP assay and results included in Figure 2A and “Appendix Figure S3”.

Concerning PPAR β , to our knowledge, it is recognized that specific commercially available PPAR β antibodies is currently lacking. It represents an important limit for studies aiming to investigate the PPAR β expression and function.

5. The manuscript is lacking the translational part. Would diabetic patients take use PPAR γ agonists benefits for lactose intolerance?

This is a really good point. Nearly 75% of patients suffering from type 2 diabetes report gastrointestinal symptoms such as flatulence, bloating, diarrhea or constipation. The significant higher frequency of lactose intolerance in type 2 diabetic patients has been suggested for the first time in a very recent prospective clinical study (SatyaVati Rana *et al. Clinica Chimica Acta* 462 (2016) 174–177). Giving that this association between type 2 diabetes and lactose intolerance has never been considered before, the potential improvement of lactose intolerance-related gastrointestinal symptoms in clinical trials testing thiazolidinediones has not been reported. For example, no such data are available in a meta-analysis published in the Cochrane Library that aimed to evaluate the effects of rosiglitazone in the treatment of type 2 diabetes (Richter *B, Cochrane Database of Systematic Reviews* 2007, Issue 3). Therefore, unfortunately, we cannot document this question in our manuscript. It seems clear that, from now, the evaluation of the lactose intolerance and its related symptoms in prospective trials aiming to assess the efficacy of PPAR γ agonist in type 2 diabetic patients would be an exciting outcome to consider.

6. The authors used shPPAR γ and in the supplementary shPPAR α . They are invited to include in the main manuscript an experiment in cells with shPPAR α , shPPAR γ , shPPAR β with and without specific agonists for the 3 receptors and detect modulation of LCT mRNA levels. This is necessary to prove the specificity of gamma receptor.

We did not use ShPPARa in the supplementary figure, it was rather the evaluation of PPARa mRNA expression in the ShPPARg cell line (previous Supplemental Figure S8). The figure was probably not enough clear and we apologize for this inconvenience. Unfortunately, the 3-month period required for paper reviewing did not allow us to establish and to use stable Caco-2 cells expressing specific ShRNA against PPARa and PPARb. The question of the specificity of PPARg-dependent regulation of LCT expression was rather investigated and confirmed by using synthetic antagonist against PPARa (GW6471). All the results concerning this specific point of PPARg specificity versus PPARa are now included in figure 2 (figure 2F, 2G and 2H). The “Results” and “Material and methods” sections were both modified in the revised version of the article to include all these new data. Considering that some of the PPARg agonists that we used are recognized to be able to slightly activate PPARa receptor (but are largely less efficient to activate PPARb), it was important to eliminate a PPARa-dependent effect of our ligands. Since our results clearly suggest that PPARa is not involved in the LCT gene expression, we can reasonably assume that PPARb is probably also not involved in the GED-, CLA-, Pio- and Rosi-dependent induction of LCT expression and activity.

We hope that revised draft of the manuscript fulfills the requirements and the suggestions of the reviewers and the editorial board.

2nd Editorial Decision

24 August 2017

Thank you for the submission of your revised manuscript to EMBO Molecular Medicine. We have now received the enclosed reports from the referees who were asked to re-assess it. As you will see, the reviewers are now supportive and I am pleased to inform you that we will be able to accept your manuscript pending the following editorial amendments:

- in M&M, please include a statement that informed consent was obtained from all subjects and that the experiments conformed to the principles set out in the WMA Declaration of Helsinki and the Department of Health and Human Services Belmont Report.

Please submit your revised manuscript within two weeks. I look forward to seeing a revised form of your manuscript as soon as possible.

***** Reviewer comments *****

Referee #1 (Remarks):

In the revised manuscript, the authors addressed most of the major points. First, the authors added new data showing that a widely used PPARgamma ligand rosiglitazone can induce LCT gene expression and activity in Caco-2 cells. Second, the quality of Western blot and quantitative experiments (qPCR and luciferase assays) improved. The authors did not perform the suggested ChIP-Seq experiment, but repeated ChIP assays including more negative controls (Figure 2A). Based on the presented data, PPARgamma appears to regulate LCT gene expression, but the claim would have been stronger if supported with unbiased PPARgamma ChIP-Seq data.

Corresponding Author Name: Benjamin Bertin

Manuscript Number: EMM-2017-07795